# Identification of a Novel Therapeutic Target against XDR *Salmonella* Typhi H58 Using Genomics Driven Approach Followed Up by Natural Products Virtual Screening

**DOI:** 10.3390/microorganisms9122512

**Published:** 2021-12-03

**Authors:** Khurshid Jalal, Kanwal Khan, Muhammad Hassam, Muhammad Naseer Abbas, Reaz Uddin, Ameer Khusro, Muhammad Umar Khayam Sahibzada, Márió Gajdács

**Affiliations:** 1International Center for Chemical and Biological Science, H.E.J. Research Institute of Chemistry, University of Karachi, Karachi 75270, Pakistan; khurshid26523@gmail.com; 2Dr. Panjwani Center for Molecular Medicine and Drug Research, International Center for Chemical and Biological Science, University of Karachi, Karachi 75270, Pakistan; khankanwal011@gmail.com (K.K.); muhammadhassam8@gmail.com (M.H.); 3Department of Pharmacy, Kohat University of Science and Technology, Kohat 26000, Pakistan; dr_naseer86@yahoo.com; 4Research Department of Plant Biology and Biotechnology, Loyola College, Chennai 600034, India; armankhan0301@gmail.com; 5Department of Pharmacy, Sarhad University of Science and Information Technology, Peshawar 25100, Pakistan; 6Department of Oral Biology and Experimental Dental Research, Faculty of Dentistry, University of Szeged, 6720 Szeged, Hungary

**Keywords:** *Salmonella* Typhi H58, subtractive genomic analysis, typhoid fever, drug targets, functional annotation, XDR

## Abstract

Typhoid fever is caused by a pathogenic, rod-shaped, flagellated, and Gram-negative bacterium known as *Salmonella* Typhi. It features a polysaccharide capsule that acts as a virulence factor and deceives the host immune system by protecting phagocytosis. Typhoid fever remains a major health concern in low and middle-income countries, with an estimated death rate of ~200,000 per annum. However, the situation is exacerbated by the emergence of the extensively drug-resistant (XDR) strain designated as H58 of *S.* Typhi. The emergence of the XDR strain is alarming, and it poses serious threats to public health due to the failure of the current therapeutic regimen. A relatively newer computational method called subtractive genomics analyses has been widely applied to discover novel and new drug targets against pathogens, particularly drug-resistant ones. The method involves the gradual reduction of the complete proteome of the pathogen, leading to few potential and novel drug targets. Thus, in the current study, a subtractive genomics approach was applied against the *Salmonella* XDR strain to identify potential drug targets. The current study predicted four prioritized proteins (i.e., Colanic acid biosynthesis acetyltransferase wcaB, Shikimate dehydrogenase aroE, multidrug efflux RND transporter permease subunit MdtC, and pantothenate synthetase panC) as potential drug targets. Though few of the prioritized proteins are treated in the literature as the established drug targets against other pathogenic bacteria, these drug targets are identified here for the first time against *S.* Typhi (i.e., *S.* Typhi XDR). The current study aimed at drawing attention to new drug targets against *S.* Typhi that remain largely unexplored. One of the prioritized drug targets, i.e., Colanic acid biosynthesis acetyltransferase, was predicted as a unique, new drug target against *S.* Typhi XDR. Therefore, the Colanic acid was further explored using structure-based techniques. Additionally, ~1000 natural compounds were docked with Colanic acid biosynthesis acetyltransferase, resulting in the prediction of seven compounds as potential lead candidates against the *S.* Typhi XDR strain. The ADMET properties and binding energies via the docking program of these seven compounds characterized them as novel drug candidates. They may potentially be used for the development of future drugs in the treatment of Typhoid fever.

## 1. Introduction

The causative agent of typhoid fever *Salmonella* Typhi is a rod-shaped flagellated Gram-negative bacterium. *S.* Typhi is a human-specific pathogenic strain. *S.* Typhi is surrounded by polysaccharides capsule that acts as a virulent factor that provides protection by deceiving the host immune system [1]. Typhoid fever remains a significant health concern in middle and low-income countries, with an estimated death of ~200,000 annually [2]. In November 2016, the XDR H58 strain was first reported in Sindh province of Pakistan, and it was responsible for approximately 5274 cases in the large cities of Sindh. Unfortunately, H58 showed resistance to available treatment options, i.e., chloramphenicol, ampicillin, and trimethoprim-sulfamethoxazole, as well as fluoroquinolones and third-generation cephalosporin, turning the typhoid fever treatment into a significant challenge [3]. According to the World Health Organization (WHO), about 20 million people are affected annually by typhoid fever worldwide (such as the UK, USA, Canada, and Iraq) in connection to Pakistan’s XDR outbreak.

The monophyletic study of *Salmonella* classified it as a newly emerged pathogenic strain associated with the H58 haplotype. Genome sequence analysis indicates the presence of resistance genes, i.e., qnrS and CTX- M-15, and designates it as highly resistant to fluoroquinolones and ceftriaxone. This strain has a greater tendency of spreading and a capability to transform from multidrug-resistant MDR to XDR [3]. Therefore, the drug-resistant strain of *S.* Typhi H58 has resulted in an urgent need for physicians to find an effective treatment since the available drugs are no more effective [4].

Often bacteria find alternative mechanisms to bypass the drug action. That is why the development of new drugs for existing drug targets is not a suitable approach for treating the disease [5]. Several potential drug targets can be investigated to cure highly pathogenic diseases by performing genome and proteome analysis. However, comparative genomic analysis is a challenging task because of its high cost. Computational approaches have eased these difficulties to some extent by data mining methods. One of the widely applicable computational approaches is the subtractive genomics that can be applied to evaluate the suitability of a new drug target based on essentiality and selectivity. This approach includes searching for new drug targets among previously identified proteins specific and essential to a particular pathogen [6,7,8]. In doing so, the following steps are involved:(i)Finding nonparalogous proteins,(ii)Finding nonhomologous to host proteins,(iii)Finding proteins that are essential for the survival of the pathogen, and(iv)Finding proteins that are playing a role in essential metabolic pathways.

This cost effect-effective has opened up new avenues for identifying possible therapeutic candidates, has sped up the drug development process, increased the number of treatment options, and lowered the rate of failure in the latter stages of clinical trials. The approach has previously been successfully employed in prioritizing and designing drug targets against *Clostridium botulinum* [9], *Mycoplasma pneumoniae* [10], *Leishmania* [11], *Rickettsia rickettsia* [12], and *Neisseria gonorrhoeae* [13].

Additionally, natural products have been widely explored as the primary source of active molecules during drug discovery. Approximately 100 new natural products are in clinical trials, particularly as anticancer and antibacterial agents. Historically, more than 80% of the drugs were associated with natural products even before the advancements in high throughput screening in the postgenomic era [14,15].

Therefore, in the current study, a subtractive genomics approach was applied to shortlist potential drug targets against the whole proteome of *S.* Typhi H58. Furthermore, natural compounds extracted from plants were also studied and classified as potent drug candidates against XDR strain. Results of this study suggested that the shortlisted proteins may act as drug targets against *S.* Typhi H58. Some novel drug candidates are also proposed to provide the basis for new drugs in the drug discovery pipeline against the H58 strain.

## 2. Materials and Methods

In the current study, a subtractive genomics analysis approach was applied against *S.* Typhi XDR strain H58 to prioritize novel drug targets [16]. The complete workflow of the current study is presented in Figure 1. Subtractive genomics is an in silico approach used to gradually reduce the whole proteome of a pathogen to a few potential and novel proteins that can act as drug targets.

### 2.1. Data Retrieval

The entire proteome of *S.* Typhi H58 was retrieved from the National Centre for Biotechnology Information (NCBI)database (www.ncbi.nlm.nih.gov/, (accessed on 17 August 2020) [17], the whole proteome of the human host was obtained from UNIPROT database (https://www.uniprot.org/proteomes/UP000005640, (accessed on 20 August 2020)), whereas the DEG (http://tubic.tju.edu.cn/deg/, (accessed on 20 August 2020)) database was retrieved to ascertain the essentiality. Moreover, data from DrugBank (https://go.drugbank.com/releases/latest, (accessed on 29 August 2020)) was utilized to ascertain the drug-target-like ability. Table 1 shows complete details of the strains used in the current study.

### 2.2. Finding the Paralogous and Nonparalogous Sequence

In order to find paralogous and nonparalogous sequences, Cluster Database at High Identity with Tolerance (CD-HIT v4.6.8) [18] was used with a sequence identity threshold of 0.8 (80%). The proteins with a sequence identity of more than 80% are considered paralogous. All the paralogous sequences were excluded, and only the nonparalogous sequences were obtained for further analysis.

### 2.3. Human Nonhomologous Proteins Identifications

The main contraindication of the recommended drug may show cross-reaction with the host proteins. Using these homologous proteins as drug targets may result in undesirable effects in patients. The obtained nonparalogous sequences were then subjected to standalone BLASTp (v2.9) using the threshold expectation value 0.0001 (E-value 10^−4^). The proteins that found sequence similarity with a human proteome were excluded from the database, and only the “No-Hit” proteins were used in the subsequent steps.

### 2.4. Determination of Nonhomologous Essential Protein

The proteins used as drug targets are essential for the pathogen’s survival. The shortlisted nonhomologous proteins from the earlier steps were compared with the Database of Essential Gene (DEG) using the BLASTp [19] with a cutoff value of 10^−5^. These essential proteins have a vital function in the survival of the pathogen, and one can use these essential proteins as drug targets against the pathogen.

### 2.5. Druggability Potential of Essential Protein

The nonhomologous essential proteins of *Salmonella* were assessed through BLASTp against Food and Drug Administration (FDA) approved proteins that act as drug targets and were retrieved from the DrugBank [20] containing different types of protein targets with their FDA approved IDs. The default parameter of E-value 10^−5^ was applied in BLASTp against DrugBank to find the novel druggable targets.

### 2.6. Identification of Host and Pathogen Metabolic Pathways

The metabolic pathways of the human host and *Salmonella* were analyzed using Kyoto Encyclopedia of Genes and Genomes (KEGG) (https://www.genome.jp/kegg/pathway.html, (accessed on 15 August 2021)) [21], metabolic pathway database, and an Automatic Annotation Server called KAAS (https://www.genome.jp/kegg/kaas/, (accessed on 3 August 2021). The metabolic pathway IDs with related information of the host and pathogen were retrieved from the KEGG database. The pathways present in both human host and pathogen were characterized as common metabolic pathways, and those found only in the pathogen were selected as unique metabolic pathways. Finally, the amino acid sequences of a protein involved only in unique metabolic pathways were retrieved from the NCBI database.

### 2.7. Identification of Subcellular Localizations

All the proteins perform a specific function in a specific location. These locations are significant, and the proteins, after synthesis, are localized to their targeted areas in the cell. The proteins, failing to transport to their specific location, may result in several diseases. All the essential nonhomologous proteins found only in unique metabolic pathways were subjected to PSORTb version 3.0.2 [22] for subcellular localization identification. The subcellular localization of proteins plays a vital role in the pathogenesis of bacteria. The main principle of subcellular localization (SCL) is to perform a BLAST search of all the essential nonhomologous proteins against the proteins of known subcellular localization. The PSORTb classified proteins based on cellular localization that involves cytoplasm, cytoplasmic membrane, periplasmic membrane, extracellular space, and unknown.

### 2.8. Homology Modeling

The PDB structure of all the shortlisted proteins was searched for if available. In the absence of the 3D structure, the protein structure can be modeled through homology modeling using phyre2 online server (http://www.sbg.bio.ic.ac.uk/phyre2/html/page.cgi?id=index, (accessed on 7 January 2021)) [23]. Homology modeling is the most accurate and reliable method for constructing the three-dimensional (3D) structure of the proteins in the absence of experimentally well-defined crystal structure. It operates based on sequence similarity with other proteins reported in the Protein Data Bank [24].

### 2.9. Modeled Structure Validation

For verification of the modeled structure, different tools were used, including ProCheck (https://servicesn.mbi.ucla.edu/PROCHECK/, (accessed on 9 January 2021)) and PsiPred (http://bioinf.cs.ucl.ac.uk/psipred/, (accessed on 9 January 2021)). The ProCheck server was used to check the stereochemical quality of a protein structure by analyzing residue-by-residue geometry and the overall structure geometry of the protein [25]. This program compares various parameters for the given protein with the ideal value obtained from the well-defined and well-refined high-resolution protein structure deposited in the PDB. However, prediction of the secondary structure (β-sheets, α-helices, and random coils) of shortlisted proteins were performed by online server Phyre2. This software used PsiPred to predict the secondary structure from the amino acid sequence. This server is a reliable tool for the secondary structure prediction from the primary sequence of the protein [23].

### 2.10. Active Site Prediction

Active site prediction was performed by standalone software Molecular Operating Environment MOE [26]. It predicts the active site of the modeled protein via three approaches:(1)Uses sequence similarities to find evolutionary conserved functional residues,(2)Structure homology with other proteins of known active site residues,(3)Uses physiochemical properties of the protein structure and function to find the active site residues [27].

### 2.11. Ligand Identification

Ligand identification of modeled apoprotein was performed by online server ProBis (Protein Binding Sites) [28]. The ProBis is the new tool used for the prediction of protein binding site and their ligand based on molecular simulation with a graph-theoretical approach. This tool compares the amino acid sequence of both query and target protein based on similarity and physiochemical properties, and as a result, it provides us with the predicted ligand. One may design new drugs in further drug discovery stages [29]. The ProBis is freely available at http://probis.cmm.ki.si, (accessed on 13 January 2021). The ProBis tool consists of a different database, algorithm, and web server for predicting the binding site of ligands on the surface of proteins, based on the structure related to similar proteins in PDB [30].

### 2.12. Molecular Docking and Virtual Screening

In the current study, molecular docking was performed by Auto Dock (v4.2) and MOE (Molecular Operating Environment). These docking tools are widely used for the prediction of protein–ligand interactions. The protein with modeled structure was used as target protein whereas 1000 natural compounds (edible plants) from the ZINC database (MolPort) (https://zinc15.docking.org/catalogs/molportnp/, (accessed on 13 June 2021)) were used as a ligand library. The standard docking procedure was applied for molecular docking using AutoDock. In addition, a redocking procedure was performed for highly docked compounds based on their binding energies and ADMET properties. The shortlisted compounds were redocked to Colanic acid biosynthesis acetyltransferase through MOE Software.

### 2.13. ADMET Analysis of Shortlisted Natural Compounds

The ADMET properties of a compound relate to the absorption, distribution, metabolism, excretion, and toxicity in and through the human body. They include the properties such as molecular weight, Caco2 permeability, blood–brain barrier percentages, plasma protein binding properties, skin permeability, compound carcinogenicity, LogP, LogD, LogSw, solubility, oral bioavailability, and Lipinski rules for drug-likeness of compounds. The ADME, drug-likeness, and toxicity properties for shortlisted natural compounds were identified through the PreADMET tool (https://preadmet.bmdrc.kr/, (accessed on 20 March 2021). The FAFDrug4 tool (https://fafdrugs4.rpbs.univ-paris-diderot.fr/, (accessed on 29 March 2021)) was used to calculate the Lipinski rules for shortlisted compounds.

### 2.14. Post-Docking Analysis/Protein–Ligand Interactions

The MOE software was used to investigate the docked ligand–protein interaction and to visualize the hydrogen bond and hydrophobic interactions of the ligand with docked protein within the range of 5Å.

### 2.15. Protein–Protein Interaction

The Protein–Protein Interactions (PPI) are the backbone of cellular machinery. We determined the Protein–Protein Interactions of shortlisted proteins via the STRING server (https://string-db.org/, (accessed on 23 February 2021)). The STRING is a database of experimentally known and predicted PPIs, including direct (physical) and indirect (functional) networks [31].

## 3. Results and Discussion

This study aims to find novel drug targets against the *S.* Typhi XDR strain H58 for the future drug discovery pipeline. These drug targets should meet the druggability criteria that include the essentiality of the protein for the survival of bacteria, being nonparalogous to the pathogen, nonhomologous to humans, and having a significant role in unique metabolic pathways of the pathogen.

### 3.1. Identification of Nonparalogous Sequence

The complete proteome of *S.* Typhi strain H58 retrieved from the NCBI database was comprised of 4501 proteins sequences. The CD-HIT predicted 37 proteins as paralogous (duplicate) out of 4501. Therefore, the paralogous sequences were excluded, and the remaining proteins were retrieved for further study.

### 3.2. Identification of Nonhomologous Proteins

The remaining 4464 proteins were subjected to BLASTp against the whole human proteome to find the nonhomologous proteins. The result showed that out of 4464 nonparalogous proteins, ~1015 proteins are homologous, having high similarity with human proteome. Thus, the resultant 3449 proteins were predicted as nonhomologous proteins. These proteins were further analyzed in subsequent steps.

### 3.3. Identification of Essential Nonhomologues Proteins

A BLASTp search of the nonhomologous proteins of *S.* Typhi against DEG was performed. The BLASTp search resulted in essential 1961 proteins responsible for vital activities in the pathogen’s life cycle and were thus considered essential. These nonhomologous essential proteins may be proposed safely as potential and pathogen-specific drug targets. In principle, by targeting such proteins, bacteria may survive, however, with less virulence, or many of its vital functions may halt, resulting in losing pathogenicity.

### 3.4. Prediction of Drug-like Ability

The 1961 nonhomologous essential proteins were then subjected to BLASTp against the DrugBank database to find the sequence similarities with drug target proteins available in this database. Only those proteins which showed significant sequence similarity with FDA-approved drug targets were selected, and the remaining were excluded from the database. Consequently, 555 proteins were shortlisted as essential, nonhomologous, and drug target-like proteins.

### 3.5. Metabolic Pathways Analysis

The nonhomologous, essential, and drug target-like proteins were then subjected to KAAS (KEGG Automatic Annotation Server) to find the proteins involved in various metabolic pathways active in the *Salmonella* H58 strain. A total of 484 proteins were predicted to be involved in various metabolic pathways crucial for the persistence of *S.* Typhi. In the following, we listed the predicted metabolic pathways in which the shortlisted proteins play significant roles (Appendix A). These roles include bacterial secretion system, two-component system, quorum sensing, bacterial chemotaxis, flagellar assembly, beta-Lactam resistance, vancomycin resistance, cationic antimicrobial peptide (CAMP) resistance, phenylalanine, tyrosine and tryptophan biosynthesis, taurine and hypo-taurine metabolism, cyano amino acid metabolism, lipopolysaccharide biosynthesis, O-antigen nucleotide sugar biosynthesis, peptidoglycan biosynthesis, pantothenate and CoA biosynthesis, polyketide sugar unit biosynthesis, monobactam biosynthesis, streptomycin biosynthesis, benzoate degradation, aminobenzoate degradation, and methane metabolism (Figure 2).

### 3.6. Prediction of Subcellular Localization of Nonhomologous Essential Proteins

In a subtractive genome analysis approach, the nonhomologous essential proteins are subjected to an online server named PSORTb to find the different localization of the proteins based on the overall amino acid composition of proteins, sequence homology, and motifs, known targeting sequence, and combined information from the above methods [32]. The protein localization is significant in the drug discovery phase because the design of new drugs and vaccines is based on protein localization. For example, cell membrane proteins are primarily used as vaccine targets, and cytoplasmic proteins are used as drug targets. In the current study, the results showed that out of 484 proteins, 309 proteins are predicted to be in the cytoplasm, 85 proteins were predicted to be localized in the cytoplasmic membrane, 36 proteins were predicted as unknown localization, 28 proteins were predicted to be localized in the periplasmic region, and 26 proteins were predicted to be localized in the outer membrane.

### 3.7. Significance of the Shortlisted Proteins

In the current study, four proteins are shortlisted as drug targets based on their functions reported in the literature [33] (Table 2). These four proteins are Colanic acid biosynthesis acetyltransferase wcaB, Shikimate dehydrogenase aroE, multidrug efflux RND transporter permease subunit MdtC, and Pantoate: beta-alanine ligase panC (Pantothenate synthetase. These four proteins may be proposed as potential drug targets due to their essential properties, being nonhomologous and nonparalogous, and involved in essential metabolic pathways (Table 3). The stepwise filtration of shortlisted proteins is shown in Figure 3.

### 3.8. Shikimate Dehydrogenase

Shikimate dehydrogenase is the enzyme involved in one step of the shikimate pathway, and it is absent in humans. It catalyzes the following reaction:(1)$$\left.\mathrm{Shikimate}+{\mathrm{NADP}}^{+}\to \;3\text{-}\mathrm{dehydroshikimate}+\mathrm{NADPH}+H^{+}\right.$$

The 3-dehydroshikimate is the precursor which participates in the biosynthesis of essential amino acid phenylalanine and tryptophan, which cannot be synthesized by humans [34]. The animals, in contrast, consume these aromatic compounds from their diet. Thus, the shikimate pathway enzymes are potential drug targets for nontoxic and antibacterial drugs [35]. For example, an inhibitor of aromatic amino acid biosynthesis pathways was reported in the literature with herbicidal activity [36,37]. It is used as a drug target to inhibit *Escherichia coli* [38], methicillin-resistant *Staphylococcus aureus* [39], *Mycobacterium tuberculosis* [40], and *Helicobacter pylori* [41].

### 3.9. Pantoate: Beta-Alanine Ligase panC (Pantothenate Synthetase)

The pantoate-beta-alanine ligase, also known as Pantothenate synthetase, catalyzes the synthesis of Pantothenate from alanine and pantoate. The Pantothenate is vitamin B5, and it is further converted to CoA in five steps reaction [42]. The CoA is a universal and essential cofactor involved in many metabolic reactions, including synthesis of phospholipids, operation of Kreb’s cycle, synthesis, and degradation of fatty acid [43]. We can use pantoate: beta-alanine ligase as a potential drug target because of its involvement in Vitamin B5 synthesis and absence in the human host. Moreover, pantothenate synthetase is one of the promising drug targets for which many inhibitors have been designed [44]. The metabolic pathways of shikimate dehydrogenase and Pantothenate synthetase can be seen in Appendix A.

### 3.10. Multidrug Efflux RND Transporter Permease Subunit Mdtc

The MdtC permease belongs to a family of RND (resistance-nodulation-division) transport. These transports are widely present in Gram-negative pathogens and actively involved in the efflux of antibiotics. These are transmembrane proteins embedded in the periplasmic domains making tripartite complex with the outer membrane. These transporters are intensively being studied in various pathogens for the efflux of a range of antibiotics. Blocking these transporters with effective drugs may lead to overcoming the antibiotic resistivity found in *Salmonella* Typhi.

### 3.11. Colanic Acid Biosynthesis Acetyltransferase Wcab

The wcaB belongs to the class of acetyltransferase responsible for the addition of acetyl group to the galactosyl residue in the UPP-linked tetrasaccharide in the Colanic acid biosynthesis [45]. The Colanic acid is the exopolysaccharide that makes a protective and pathogenic capsule to *Salmonella* [[46,47]. The Colanic acid enables the bacteria to survive in the acidic environment of the gastrointestinal tract [48]. The inhibition of the Colanic acid biosynthesis acetyltransferase will halt the synthesis of Colanic acid. The bacteria cannot survive in an acidic environment. Thus, it is proposed as a potential drug target. It performs the following reaction:(2)$$\begin{array}{c} \beta \text{-}D\text{-}\mathrm{galactosyl}\text{-}\left(1\to 3\right)\text{-}\alpha \text{-}L\text{-}\mathrm{fucosyl}\text{-}\left(1\to 4\right)\text{-}\left[2/3\text{-}O\text{-}\mathrm{acetyl}\text{-}\alpha \text{-}L\text{-}\mathrm{fucosyl}\right]\text{-}\left(1\to 3\right)\text{-}\;\\ \left.\beta \text{-}D\text{-}\mathrm{glucopyranosyl}\text{-}\mathrm{diphosphoundecaprenol}+\mathrm{acetyl}\text{-}\mathrm{CoA}⟶\;2\text{-}O\text{-}\mathrm{acetyl}\text{-}\beta \text{-}D\text{-}\mathrm{galactosyl}\text{-}\left(1\to 3\right)\text{-}\right.\;\\ \alpha \text{-}L\text{-}\mathrm{fucosyl}\text{-}\left(1\to 4\right)\text{-}\left[2/3\text{-}O\text{-}\mathrm{acetyl}\text{-}\alpha \text{-}L\text{-}\mathrm{fucosyl}\right]\text{-}\left(1\to 3\right)\text{-}\beta \text{-}D\text{-}\mathrm{glucopyranosyl}\text{-}\mathrm{diphosphoundecaprenol}\;\;\\ +\mathrm{coenzyme}\;A \end{array}$$

### 3.12. Homology Modeling of Shortlisted Drug Target

The 3D structure of Colanic acid biosynthesis acetyltransferase (one out of four shortlisted) was unavailable in the PDB. Therefore, its homology modeling was performed by taking the FASTA sequence of the protein from the NCBI database with the accession number WP_000888724.1 as mentioned in the database. The PDB structures 1T3D, 1SSQ, and 4N6B were found as potential templates with percent identities 35%, 31%, and 30%, respectively. Eventually, the structure 1T3D was selected due to its high similarity as a template and successfully modeled the structure, as shown in Figure 4.

### 3.13. 3D Structure Validation through Ramachandran Plot (PROCHECK)

The ProCheck results showed that 81.6% residues lie in the most favored regions, 12.1% residues lie in the additional allowed regions, whereas 4.3% residues lie in the generously allowed region, and 2.1% residues were placed in disallowed regions, as shown in Appendix A. The ProCheck passed the modeled structure with statistically improved analysis, as mentioned above.

### 3.14. Secondary Structure Validation

The PsiPred results revealed that the secondary structure of *Salmonella* Typhi’s Colanic acid biosynthesis acetyltransferase consisted of a high number of beta-sheets than alpha-helices, as shown in Appendix A. The PsiPred results verified the protein based on their sequence for the alpha helices and beta sheets formation. The predicted secondary structure is represented by different signs and colors. For example, green helices represent α-helices, blue arrows indicate β-strands, and faint lines indicate coil. The SS lines represent the prediction confidence of the secondary structure. The red to blue show high to low confidence in the prediction. The disordered region is represented by question marks.

### 3.15. Active Site Prediction

The prediction of a protein’s active site is vital for various bioinformatics applications such as structure-based drug discovery and molecular docking studies. The active site was predicted by comparing the targeted protein with similar proteins sharing the known active sites. It resulted in different active sites with functional residues, as shown in Table 4. We selected the first active site based on high energy highlighted by the red color (Figure 5).

### 3.16. Ligand Identification

The ProBis result showed that the protein has high similarity with Soybean Serine Acetyltransferase Complexed with Co-enzyme A (CoA) (PDB ID: 4N6B). The identification of protein binding site and their corresponding ligands have an intensive role in drug target identification and drug research. The protein binding sites are structurally and functionally important regions on the protein surface on which different types of drugs interact to perform the desired action [28]. Numerous computational tools (i.e., CASTp, LigASite, PDBeMotif, 3DLigandStie, Sites Base, etc.) are used for the prediction and analysis of ligand binding sites, as reported in the literature [49]. The identified ligand CoA has helped in the next stage of molecular docking study for the redocking experiment.

### 3.17. Molecular Redocking Study

We performed the docking study of Colanic acid biosynthesis acetyltransferase using AutoDock 4.2. The ligand was docked using ten algorithm runs with setting Lamarckian GA for 250 times along with 2,500,000 maximum number of evaluation steps resulting in 27,000 number of generations. The AutoDock results showed the binding of the ligand in the active site of protein with different conformations and orientations. Every conformation showed different binding energy from negative to positive. The top-ranked conformation was selected based on the lowest binding energy (i.e., −6.84 kcal/mol) since the conformation with the lowest binding energy corresponds to the fact that the binding of the ligand with the active site is spontaneous, and also it forms a lower energy complex (i.e., more stable). The redocked compound with the modeled protein is shown in Figure 5.

### 3.18. Post Redocking Analysis (Redocking)

The post docking analysis was performed with the help of MOE software. The results showed that the active site residues Gln61 mediates hydrogen bond with OH group of Ribose, Asn37 makes a hydrogen bond with NH group, Asn81 formed two hydrogen bonds with two oxygen atoms of the phosphate group, Arg49, Lys82, and Asn100 mediate hydrogen bonds with seven oxygen atoms of three phosphate groups, Ala42 make one hydrogen bond with OH group and another hydrogen bond with carbonyl oxygen, and Trp40 mediates a hydrogen bond with carbonyl oxygen.

### 3.19. Virtual Screening

The ZINC library of natural products (from an edible plant source) was docked against the modeled structure. The natural product database was screened for the number of compounds having a molecular weight ranging from 150 to 350 and resulted in one thousand compounds that were later used in virtual screening. These natural products from edible plants were used because of their easy availability. The compounds were docked by the same parameters as used previously for the AutoDock validation (redocking). In the case of Colanic acid biosynthesis acetyltransferase, only 380 compounds showed favorable interactions with energetics ranging from −7.2 to −11.2 kcal/mol, as shown in Appendix A.

### 3.20. Docking of Shortlisted Natural Compounds

From the above virtual screening, seven compounds, i.e., Acetoxy-(10)-gingerol, 1-(4-Hydroxy-3-methoxyphenyl)-3,5-diacetoxyoctane, (6)-Gingerdiol acetate methyl ether, Feruloylcholine,4-(3-Hydroxy-7-phenyl-6-heptenyl)-1,2-benzenediol, Tetramethylquercetin, and Isosalsolidine were shortlisted as lead drug molecule against Colanic acid biosynthesis acetyltransferase based on their ADME, toxicity, drug-likeness and Lipinski rules predicted through PreADMET, and FAFDrug4 tool. These seven shortlisted natural product compounds are the main constituents of edible plants such as ginger, garlic, turmeric, oregano, etc. The identified gingerol compounds ((6)-Gingerdiol acetate methyl ether, Acetoxy-(10)-gingerol) are traditionally used [50] for the treatment of cancers, dyspepsia and are widely evaluated against *Bacillus* spp., *Escherichia coli*, *Staphylococcus aureus*, and *periodontal* bacteria for their antimicrobial activities [51,52]. Moreover, Feruloylcholine (having a curcumin moiety) is also reported for its antimicrobial activity against Gram-positive (*S. aureus* and *B. subtilis*) and Gram-negative (*S. Typhi* and *K. pneumoniae*) [53]. Additionally, Silva et al. recently reported the antifungal activity of Tetramethylquercetin [54]. These shortlisted seven compounds were subjected to molecular docking through MOE for detailed study, and it showed binding energies ranging from ~−5 to −7 Kcal/mol. The hydrogen bonding and hydrophobic interactions with Colanic acid are shown in Figure 6. The shortlisted compounds, along with the binding energies, were presented in Table 5. The ADMET properties of these shortlisted compounds were presented as Appendix A. All the parameters were found in favor of the drug-like characteristics of the shortlisted seven compounds.

### 3.21. Protein–Protein Interaction and Functional Annotation

Many functional and physical interactions occur between different types of proteins and are crucial for many biological processes of cellular machinery. The data on these interactions can be used to filter and evaluate functional genomic data for annotating functional, structural, and evolutionary information on the proteins. Investigating the predicted PPIs may offer a new direction for future experimental research and computer-aided drug discovery [55]. The STRING server was used to determine PPI and functional annotation of shortlisted proteins (i.e., Colanic acid biosynthesis acetyltransferase wcaB, Peptidoglycan glycosyltransferase MrdB, Shikimate dehydrogenase aroE, multidrug efflux RND transporter permease subunit MdtC). The STRING results showed different nodes and edges of each protein. The number of nodes represents proteins such as splice isoforms and post-translational modified proteins. Each node represents a specific protein encoded by a single gene. Node color tells us about the protein interaction, a colored node represents query protein and first shell of interactors, a white node represents the second shell of the interactor, an empty node represents the protein of unknown 3D structure, and a filled node represents proteins having a 3D structure or predicted structure. Furthermore, edges represent the physical and functional protein–protein associations. The color of the line edges represents the different sources of interactor protein; for example, cyan color represents the protein from a curated database, violet color represents experimentally determined protein, green color represents neighborhood gene, red color represents fusion gene, lemon color represents text mining source, black color represents the co-expressed proteins, and blue color represents homologous proteins [31]. The results also showed that the prioritized target proteins might act as hub proteins interacting with more than ten proteins. Therefore, targeting such proteins may affect the activity of all interactor proteins.

#### 3.21.1. ColanicAcid Biosynthesis Acetyltransferase Wcab

The STRING database was uploaded with the NCBI ID: WP_000888724.1 of Colanic acid biosynthesis acetyltransferase and found the interaction with other proteins in the *Salmonella* Typhi. The Colanic acid biosynthesis acetyltransferase was represented by wcaB, and its interactions with other proteins present in their neighbor are wcaC (annotation not available for this protein), t1553 (Pyruvate-flavodoxin oxidoreductase), cysK (Cysteine synthase A), metH (annotation not available), wzc (Tyrosine-protein kinase), relE (annotation not available), yfiF(annotation not available), trmH (tRNA (guanosine(18)-2′-O)-methyltransferase), rlmB (23S rRNA (guanosine-2′-O-)-methyltransferase), and cysM (annotation not available). The results showed that the Colanic acid biosynthesis acetyltransferase (wcaB) protein has a total of 18 edges, the number of expected edges was 13, the number of nodes 11, and its average nodes degree was 3.27. The Protein–Protein Interaction enrichment *p*-value is 0.123 with an average local clustering coefficient of 0.868 (Figure 7A). These proteins are involved in different vital activities. Therefore, by targeting wcaB protein, the rest of the interacting proteins may also lose their function. Thus, we may safely propose this protein as a drug target.

#### 3.21.2. Shikimate Dehydrogenase AroE

Similarly, the Shikimate dehydrogenase with accession number WP_000451199.1 was uploaded to the STRING database. The results showed many interactions with *Salmonella* Typhi family proteins. The Shikimate dehydrogenase was represented by aroE having interactions with arok (Shikimate kinase), aroL (Shikimate kinase 2), aroD (3-dehydroquinate dehydratase), aroB (3-dehydroquinate synthase), t4102 (Threonyl carbamoyl-AMP synthase), t6434 (annotation not available), yrdB (annotation not available), aroA (3-phosphoshikimate 1-carboxyvinyltransferase), yrdD (annotation not available), and aroC (Chorismate synthase). The STRING results showed that Shikimate dehydrogenase has a total number of nodes: 11, number of edges 31, expected number of edges: 10, average node number: 5.64, with average local clustering coefficient of 0.9, PPI enrichment *p*-value is 2.01 × 10^−7^ (Figure 7B). The Shikimate dehydrogenase is also involved in various other pathways such as aromatic amino acid family biosynthetic process, chorismate biosynthetic process, and shikimate metabolic process. Hence, we may safely propose Shikimate dehydrogenase as a drug target.

#### 3.21.3. Multidrug Efflux RND Transporter Permease Subunit Mdtc

Similarly, the multidrug efflux RND transporter permease subunit MdtC having NCBI ID: WP_001210089.1 was uploaded to the STRING database and determined the interactions with another protein family of *Salmonella*. The result showed various interactions of Mdtc with other proteins such as MdtA (multidrug-resistant protein), MdtD (putative multidrug resistance protein), DD95_00220 (Two-component system sensor histidine kinase BaeA), DD95_00215 (Response regulator in a two-component regulatory system with BaeS), MdtB (multidrug resistance protein), EmrB (multidrug efflux MFS transporter subunit), emrD (multidrug efflux protein involved in adaptation to low energy shock), DD95_16625 (multidrug transporter), DD95_16615 (MexE family multidrug efflux RND transporter periplasmic adaptor subunit), and DD95_07805 (MexE family multidrug efflux RND transporter periplasmic adaptor subunit). The STRING result showed that MdtC protein formed 11 nodes, 31 edges, average node degree was 5.64, avg local cluster coefficient 0.825, the expected number of edges was 10, and PPI enrichment *p*-value was 1.21e-07 (Figure 7C). This protein is involved in the export of many drugs (antibiotics) out of the cell, so the antibiotics are unable to reach their final target to inhibit their target enzyme.

#### 3.21.4. Pantoate: Beta-Alanine Ligase panC (Pantothenate Synthetase)

Similarly, the Pantoate: beta-alanine ligase panC (pantothenate synthetase) NCBI ID: WP_000706998.1 was uploaded on the STRING database, and the direct as well as indirect interactions of this protein with neighbor proteins in *Salmonella* Typhi H58, was found. The STRING result showed many interactions of panC with panD (Aspartate 1-decarboxylase), panB (3-methyl-2-oxobutanoate hydroxy methyltransferase), panE (2-dehydropantoate 2-reductase), t0284 (2-dehydropantoate 2-reductase), coaA (annotation not available), dfp (Coenzyme A biosynthesis bifunctional protein CoaBC), gabT (annotation not available), cmk (annotation not available), birA (Bifunctional ligase/repressor BirA), and coaD (Phospho-pantetheine adenylyl transferase). The STRING result showed that panC had 11 nodes, 26 edges, the average node degree was 4.73, average local clustering coefficient 0.717, the expected number of edges was 11, and PPI enrichment *p*-value was 4.49e-05 (Figure 7D). This enzyme is involved in beta-alanine metabolism and pantothenate and CoA biosynthesis.

## 4. Conclusions

Typhoid fever remains a distressing public health concern inflicting the health burden worldwide. The comparative analysis showed that the genome makeup of salmonella permits its adaptation to various environmental stresses and raises difficulties in the treatment options. The genome and proteome of many pathogens are available in various open-access databases, thus speeding up the prediction of drug targets. In the current study, a subtractive genomic analysis was employed to demonstrate valuable information relevant to predicting and identifying the nonhomologous essential druggable proteins against *Salmonella* Typhi H58 as potential drug targets.

Implementing alternative strategies to combat infections seems necessary due to the evolution and rapid development of new antibiotic-resistant genes in bacterial pathogens. The availability of pathogens and host genomes allowed bioinformatics approaches to become more efficient, time-effective, and less costly. The comparative subtractive genomics associated with an analysis of metabolic pathways efficiently contributes to identifying nonhost homologous proteins that are essential to the pathogen. The subtractive genomics approach of the current study revealed three proteins of *Salmonella* Typhi H58 as novel drug targets, i.e., Shikimate dehydrogenation, Colanic acid biosynthesis acetyltransferase, Pantothenate synthetase, and multidrug efflux RND transporter protein Mdtc. Quiroz et al., as well as Mendiola et al., reported the inhibition of shikimate dehydrogenase as a potential drug target to inhibit *E. coli* and MRSA, whereas other studies have reported it inhibiting the pantothenate synthetase against MTB to minimize the prolonged TB treatment [44]. However, Jebastin et al. and Kazi et al., described the role of outer membrane proteins such as Emrs, OMPs, and Mdts as a potent vaccine candidate for chimeric vaccine designing against *Shigella flexneri* [56] as well as *Salmonella* Typhi [57]. Similarly, the identified Mdtc protein can be considered for future vaccine design against *S*. Typhi. Inhibition of these proteins may be helpful to combat the infectious diseases caused by *Salmonella* Typhi since these proteins are involved in the pathogen-specific metabolic pathways. Moreover, the identified proteins are highly conserved within the *Salmonella* Typhi strains and other pathogens (i.e., MTB, *H. pylori*, *Shigella* species, etc.) and can, therefore, be utilized as a potent drug target against other pathogens as well. Additionally, the prioritized seven natural compounds may be investigated for the inhibition of such deadly pathogens. Furthermore, chances of cross-reactivity between drugs and human proteins are reduced since the proposed drug targets showed ‘no similarity ‘with the human host proteome and, therefore, should not be considered as ‘antitargets’.

However, all the rest of nonhomologous essential proteins may also act as promising drug targets. Targeting these proteins’ functions through new drug candidates, vaccines, and therapeutics may eventually lead to the destruction and the eradication of pathogens from their host. The analysis and results of this study covered all essential and potent drug targets in *Salmonella* H58 and may help facilitate future researchers in developing effective drugs and vaccines. Furthermore, the current study identified a novel set of natural products from edible plant sources that may have the potency to inhibit *S*. Typhi. It also provided sufficient PPIs network characteristics that may be of further use in predicting novel drug targets and drug repurposing by combining various computational designing strategies for *Salmonella* Typhi H58 and other highly pathogenic bacteria.

## Figures and Tables

**Figure 1 microorganisms-09-02512-f001:**
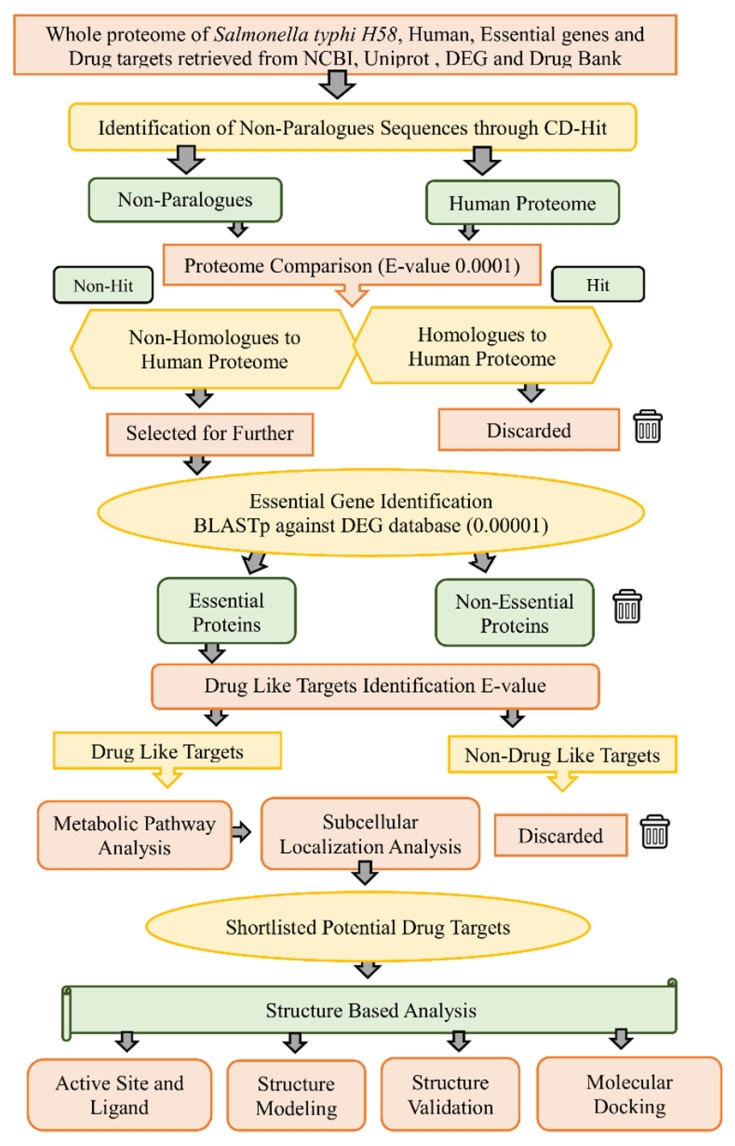
Workflow. Complete work flowchart of the present study for subtractive genome analysis of H58 for finding a potential drug target.

**Figure 2 microorganisms-09-02512-f002:**
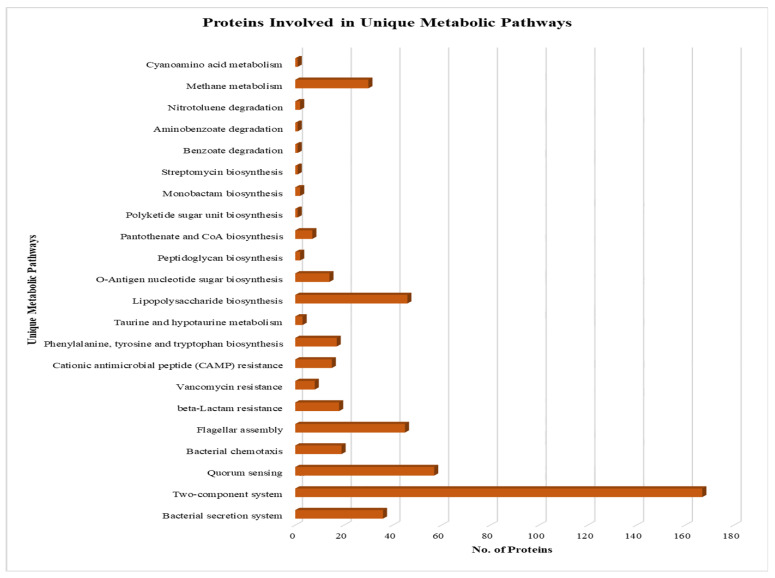
Metabolic Pathways of shortlisted Proteins. A bar showing all the proteins involved in metabolic pathways.

**Figure 3 microorganisms-09-02512-f003:**
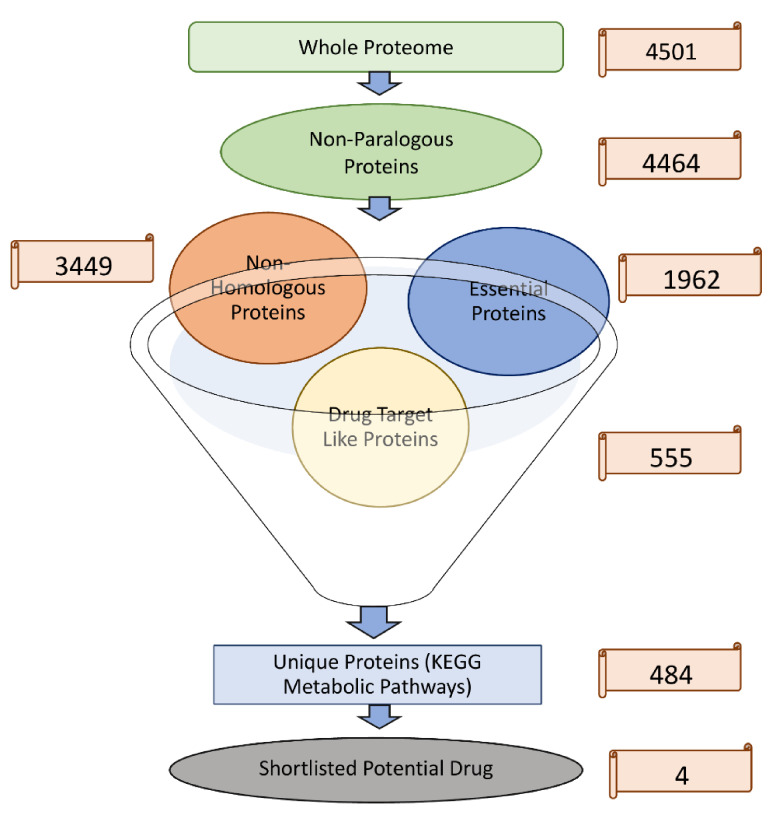
Stepwise shortlisting of proteins. Unique proteins involve in unique metabolic pathways.

**Figure 4 microorganisms-09-02512-f004:**
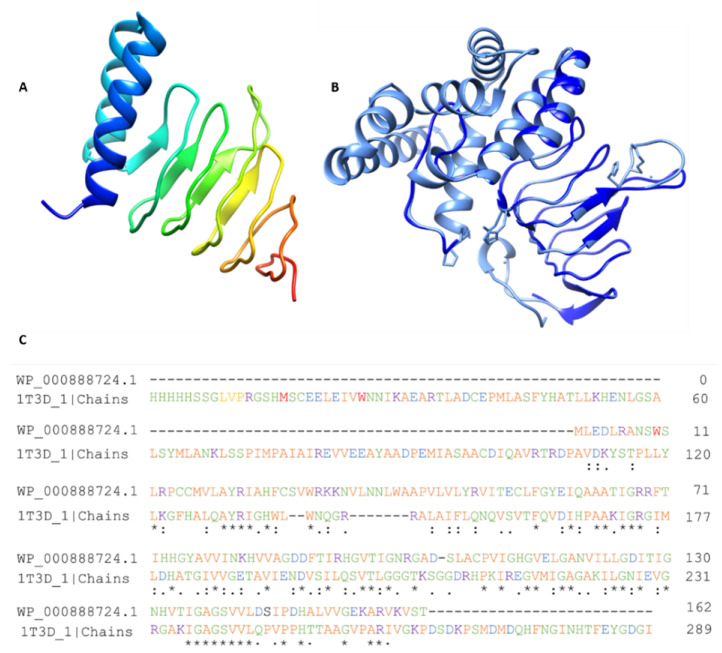
Homology modeling. (**A**) Modeled structure of Colanic acid biosynthesis acetyltransferase, (**B**) Modelled structure, and (**C**) alignment of the template and desire protein.

**Figure 5 microorganisms-09-02512-f005:**
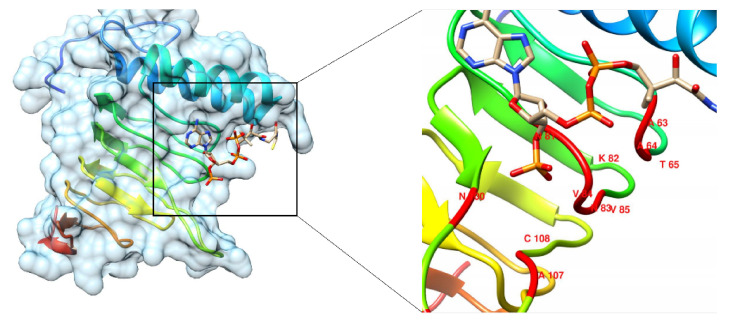
Molecular Docking. Docked conformation of ligand and protein complex.

**Figure 6 microorganisms-09-02512-f006:**
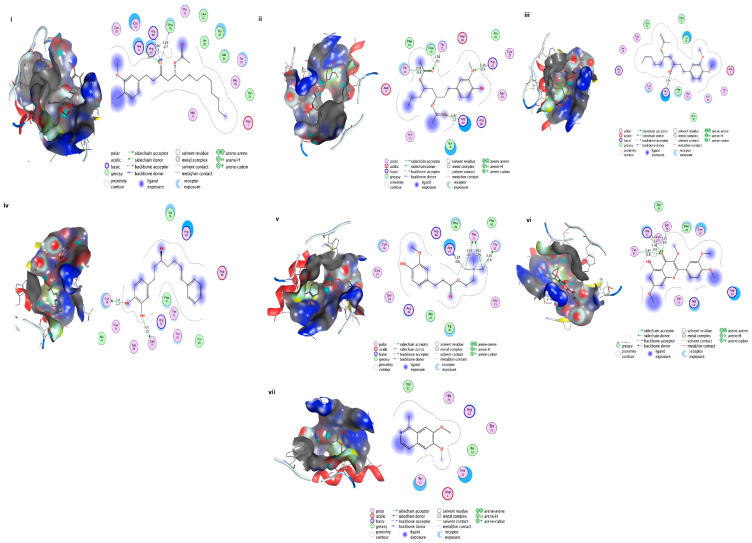
Shortlisted Compounds Interaction: Interaction of 7 shortlisted compounds with Colanic acid biosynthesis acetyltransferase (**i**) Acetoxy-(10)-gingerol, (**ii**) 1-(4-Hydroxy-3-methoxyphenyl)-3,5-diacetoxyoctane, (**iii**) (6)-Gingerdiol acetate methyl ether, (**iv**) Feruloylcholine, (**v**) 4-(3-Hydroxy-7-phenyl-6-heptenyl)-1,2-benzenediol, (**vi**) Tetramethylquercetin, (**vii**) Isosalsolidine.

**Figure 7 microorganisms-09-02512-f007:**
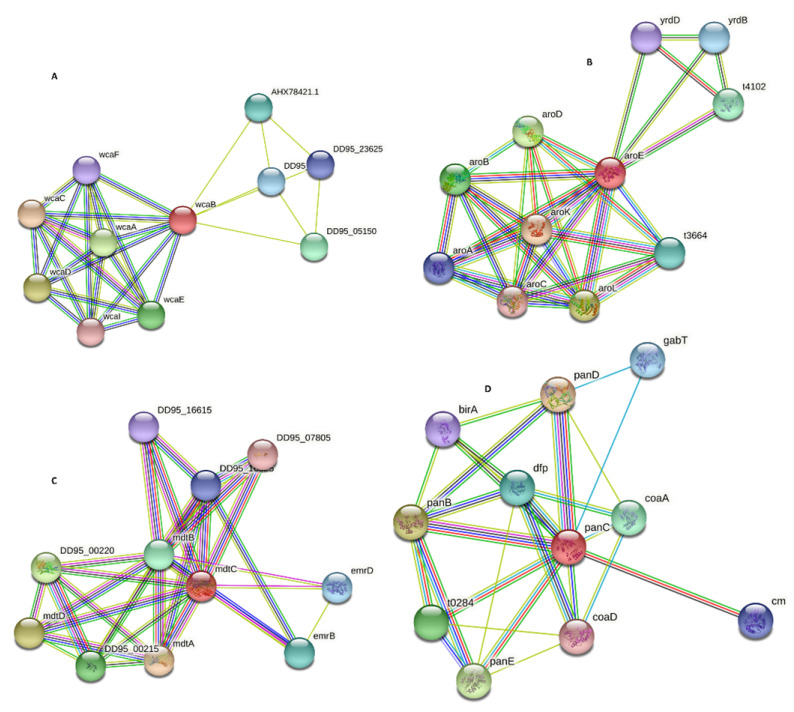
PPIs of Colanic acid (**A**), Shikimate Dehydrogenase (**B**), MdtC Protein (**C**), and panC Protein (**D**). Protein-protein interaction of identified novel drug targets.

**Table 1 microorganisms-09-02512-t001:** Complete Proteome of Human-Host and Salmonella.

Strain ID	Strain Name	Proteins
GCF_900185485.1	H58	4501
9606	Human	~200,000

**Table 2 microorganisms-09-02512-t002:** Shortlisted proteins in the current study.

Serial No.	Protein Name	Protein IDs
1	Pantoate: beta-alanine ligase	WP_000706998.1
2	Shikimate dehydrogenase	WP_000451199.1
3	Colanic acid biosynthesis acetyltransferase	WP_000888724.1
4	Multidrug efflux RND transporter permease subunit	WP_001210089.1

**Table 3 microorganisms-09-02512-t003:** Drugs target shortlisted proteins.

Serial No.	Name	DrugBank Targets	Protein ID	Ligand DrugBank ID
1	Colanic acid biosynthesis acetyltransferase	Serine acetyltransferase	Q0WKM4	DB02078
		Serine acetyltransferase	P43886	DB01992
		Galactoside o-acetyltransferase	P07464	DB01862
2	Shikimate dehydrogenase	Shikimate dehydrogenase	P15770	DB03461; DB04447
		Shikimate dehydrogenase	P43876	DB02363
3	Pantoate-beta-alanine ligase	Pantothenate synthetase	Q5SHF5	DB03570
		Pantothenate synthetase	P0A5R0	DB01930; DB02596
4	Multidrug efflux RND transporter permease subunit	Acriflavine resistance protein B	P31224	DB03825

**Table 4 microorganisms-09-02512-t004:** Active site Amino acids.

Site	Energy	Amino Acids
1	1.34	Ala63, Ala64, Thr65, Asn81, Lys82, His83, Val84, Val85, Asn100, Ala103, Asp104, Ala107, Cys108
2	0.74	Tyr21, Arg22, Ile23, Ala24, Phe70, Thr71, Ile72, His73, his74, Gly75
3	0.22	Gly114, Val115, Glu116, His132, Val133, Thr134, His148, Thr162
4	−1.13	Trp30, Asn34, Ala64, Thr65, Ile66, Gly76, Ala86, Gly87
5	−1.16	Ala24, Cys27, Ser28, Arg31, Arg69, phe70

**Table 5 microorganisms-09-02512-t005:** Docked energies of shortlisted nine natural compounds.

S. No	Compound Name	Structure	Binding Energies (Kcal/mol) MOE Docking Result
1	Acetoxy-[10]-gingerol	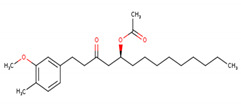	−7.21
2	1-(4-Hydroxy-3-methoxyphenyl)-3,5-diacetoxyoctane	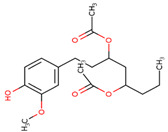	−6.39
3	[6]-Gingerdiol acetate methyl ether	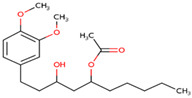	−6.65
4	Feruloylcholine	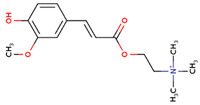	−5.8
5	4-(3-Hydroxy-7-phenyl-6-heptenyl)-1,2-benzenediol	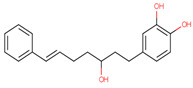	−5.92
6	Tetramethylquercetin	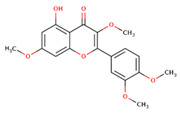	−5.88
7	Isosalsolidine	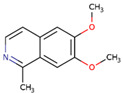	−4.77

## Data Availability

All data generated or analyzed during this study are included in this published article (and its Appendix A files).

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
