# Peer review of "Identification of a Novel Therapeutic Target against XDR Salmonella Typhi H58 Using Genomics Driven Approach Followed Up by Natural Products Virtual Screening"

_microorganisms, 2021, doi:10.3390/microorganisms9122512_

Round 1
Reviewer 1 Report
The paper by Jalal et al. entitled “Identification of a Novel Therapeutic Target against XDR Salmonella
typhi H58 Using Genomics Driven Approach followed up by Natural Products
Virtual Screening” reports a computational method, the subtractive genomic analysis to discover new target against Salmonella XDR strain. The study identified four proteins, in particular wCAb predicted as unique new drug target against Salmonella XDR strain. The docking of natural products with wCAb resulted in the prediction of seven potential lead compounds.
The paper is clear in the description of the methods and results obtained and provide a novel study to find new targets against Salmonella thypi.
Since the authors state that this approach based on subtractive genomic is relatively new, I would suggest to provide information about previous use of this kind of approach in the prediction /identification of new lead compounds against for example other types of microorganisms.
Has this approach identified new drugs used as therapeutic treatments against bacterial infections?
Do you plan to develop/synthetize these lead compounds for in vitro/in vivo biological tests?
Author Response
The paper by Jalal et al. entitled “Identification of a Novel Therapeutic Target against XDR Salmonella typhi H58 Using Genomics Driven Approach followed up by Natural Products Virtual Screening” reports a computational method, the subtractive genomic analysis to discover new target against Salmonella XDR strain. The study identified four proteins, in particular wCAb predicted as unique new drug target against Salmonella XDR strain. The docking of natural products with wCAb resulted in the prediction of seven potential lead compounds.
ï‚· The paper is clear in the description of the methods and results obtained and provide a novel study to find new targets against Salmonella typhi.
Author Response: We are thankful to the reviewer for appreciating our work.
Since the authors state that this approach based on subtractive genomic is relatively new, I would
suggest to provide information about previous use of this kind of approach in the prediction /identification
of new lead compounds against for example other types of microorganisms.
Author Response: We are thankful to the reviewer for such careful review and constructive suggestions. We have provided some examples of such approach (subtractive genomic) used for numerous pathogens from literature and also incorporated in introduction section (Second last paragraph).
ï‚· Has this approach identified new drugs used as therapeutic treatments against bacterial
infections?
Author Response: There are many literature examples where this approach paved the ways of finding the
unique drug targets against many bacterial infections as reported in introduction section.
ï‚· Do you plan to develop/synthetize these lead compounds for in vitro/in vivo biological tests?
Author Response: We have a limited lab for computational studies. However, we keep collaborating with the experimental groups and share our in silico results. Such collaborations have resulted in meaningful outcomes. We will definitely collaborate with the experimental groups and publish the findings separately.
Reviewer 2 Report
In My opinion The manuscript can be accepted, I have no comments
Author Response
In My opinion The manuscript can be accepted, I have no comments
Author Response: We are thankful to the reviewer for accepting our work.
Reviewer 3 Report
This paper by Jalal et. al. aims to find novel drug targets for Salmonella enterica serovar Typhi H58 using in silico approaches of reverse genetics and virtual screening. They shortlist four proteins as potential drug targets and find seven natural products that virtually dock on one of these proteins. Overall the relevance of the reverse genetics approach is hard to determine since the four shortlisted proteins were chosen based off literature searches. A lot more discussion of the findings (the four proteins, the seven drug candidates) is required to highlight what the unique findings of this study are.
Major points:
- The point of the subtractive genomics analysis is hard to determine. From a start of 4502 proteins the analysis reduces this to 484. At this point four proteins are then chosen to focus on based on literature? What was the value add of the subtractive genomics?
- The authors need to be clear with their definition of essential- as many of the functional groups seem to be non-essential for growth. Therefore is seems essential is used here as ‘bacteria won’t be as virulent’.
- The use of ‘metabolic pathways’ in 3.5 is wrong- this is functional groups. Many of these groups are not involved in metabolic pathways. The figure itself is hard to interpret with small colours in legend, overlapping words and very similar colours used in pie chart. No supplementary table listing the proteins investigated.
- The four potential targets identified are not discussed in context enough. How conserved are these proteins in Salmonella species in general? In proteobacteria? There is clearly literature on some of these as potential drug targets in other species but this is not discussed at all. What makes this a novel finding for this strain in particular? Would drugs to these proteins target a wide range of bacterial species?
- The in silico drug screen identified seven potential natural compounds for potential drugs (Table 5). None of these compounds are discussed for current literature around their anti-bacterial potential- such as PMID: 18814211.
- Page 17 suddenly claims MdtC is a vaccine target, not drug target with no supporting evidence?
- Overall not enough discussion of the findings, or why this is novel for this particular strain.
Minor points
- Grammar throughout entire article needs improving.
- Salmonella typhi is incorrect. It is Salmonella enterica serovar Typhi- thereafter should be shortened to Salmonella Typhi or Typhi (note italics)
Author Response
Major points:
ï‚· The point of the subtractive genomics analysis is hard to determine. From a start of 4502 proteins
the analysis reduces this to 484. At this point four proteins are then chosen to focus on based on
literature? What was the value add of the subtractive genomics?
Author Response: The subtractive genomics brought about the due attention to those proteins which have never been proposed as rug targets before or largely ignored due to less experimental evidence. In fact, no one paid attention to consider them ad drug targets. Sometime, the shortlisted proteins are the
ones that are hypothetical proteins and never considered as drug targets. However, the bacteria secrets new and unique proteins depending on its environment and those new proteins are prioritized by applying
the subtractive genomics. This is how the subtractive genomics plays a vital role to find new drug targets to combat the drug resistance.
ï‚· The authors need to be clear with their definition of essential- as many of the functional groups
seem to be non-essential for growth. Therefore, is seems essential is used here as ‘bacteria
won’t be as virulent’.
Author Response: We are thankful identifying this. The correction is made in revised manuscript.
ï‚· The use of ‘metabolic pathways’ in 3.5 is wrong- this is functional groups. Many of these groups are not involved in metabolic pathways. The figure itself is hard to interpret with small colours in legend, overlapping words and very similar colours used in pie chart. No supplementary table listing the proteins investigated.
Author Response: Agree to this point up to certain extent. However, the term ‘metabolic pathways’ is associated to the KAAS server which in turn is used to browse the KEGG database. This browsing resulted in finding those metabolic pathways in which the filtered proteins take the role. The names of such pathways are given in the section 3.5. Nevertheless, as suggested the figure is now changed in revised manuscript with clear interpretations. The detail of the proteins involved in these pathways are provided in Supplementary Data S1.
ï‚· The four potential targets identified are not discussed in context enough. How conserved are these proteins in Salmonella species in general? In proteobacteria? There is clearly literature on some of these as potential drug targets in other species but this is not discussed at all. What makes this a novel finding for this strain in particular? Would drugs to these proteins target a wide
range of bacterial species?
Author Response: Thanks for identifying this. As per suggestions, the respective parts are modified (in results under “Significance of the Shortlisted Proteins”) and conclusion section.
ï‚· The in-silico drug screen identified seven potential natural compounds for potential drugs (Table 5). None of these compounds are discussed for current literature around their anti-bacterial potential- such as PMID: 18814211.
Author Response: The manuscript is now revised according to the reviewer suggestion.
ï‚· Page 17 suddenly claims MdtC is a vaccine target, not drug target with no supporting evidence?
Author Response: We are thankful to the reviewer for identifying this. We have added remarks for these
findings in conclusion section.
ï‚· Overall not enough discussion of the findings, or why this is novel for this particular strain.
Author Response: Thank you for making this suggestion. In the revised manuscript, we have discussed the results in details.
Minor points
ï‚· Grammar throughout entire article needs improving.
Author Response: We are thankful to the reviewer for appreciating our work. We have considered all the comments from all the reviewers and editor, and have acted accordingly. The manuscript has extensively edited for language corrections, and the revised version is corrected as per the standard grammar.
ï‚· Salmonella typhi is incorrect. It is Salmonella enterica serovar Typhi- thereafter should be shortened to Salmonella Typhi or Typhi (note italics)
Author Response: We are thankful to the reviewer for identifying this mistake. We have corrected this in revised manuscript.
Round 2
Reviewer 3 Report
The authors have responded adequently to most of my comments made above however I still two outstanding issues:
1) Figure 2 is still hard to interpret with overlapping words and the colour boxes in legends too small to determine.
2) Still come places grammer could be improved in introduction but improved in general.
